# Adaptive Cellular Immunity against African Swine Fever Virus Infections

**DOI:** 10.3390/pathogens11020274

**Published:** 2022-02-20

**Authors:** Alexander Schäfer, Giulia Franzoni, Christopher L. Netherton, Luise Hartmann, Sandra Blome, Ulrike Blohm

**Affiliations:** 1Friedrich-Loeffler-Institut, Federal Research Institute for Animal Health, 17493 Greifswald-Insel Riems, Germany; alexander.schaefer@fli.de (A.S.); luise.hartmann@fli.de (L.H.); sandra.blome@fli.de (S.B.); 2Department of Animal Health, Istituto Zooprofilattico Sperimentale della Sardegna, 07100 Sassari, Italy; giulia.franzoni@izs-sardegna.it; 3The Pirbright Institute, Ash Road, Pirbright, Woking, Surrey GU24 0NF, UK; christopher.netherton@pirbright.ac.uk

**Keywords:** African swine fever virus, adaptive immunity, T cells, antigen-presenting cells, cytokines, vaccines, porcine disease

## Abstract

African swine fever virus (ASFV) remains a threat to global pig populations. Infections with ASFV lead to a hemorrhagic disease with up to 100% lethality in Eurasian domestic and wild pigs. Although myeloid cells are the main target cells for ASFV, T cell responses are impacted by the infection as well. The complex responses remain not well understood, and, consequently, there is no commercially available vaccine. Here, we review the current knowledge about the induction of antiviral T cell responses by cells of the myeloid lineage, as well as T cell responses in infected animals, recent efforts in vaccine research, and T cell epitopes present in ASFV.

## 1. African Swine Fever Virus

Over the past few years, African swine fever (ASF) has steadily increased in importance, and its spread has reached pandemic proportions [1]. In addition to Africa, where the virus has its roots in a sylvatic cycle involving warthogs and soft ticks [2], many countries in Europe and Asia are now affected, and recently the first countries in the Americas have also reported outbreaks (OIE WAHIS web interface, visited 5 January 2022). Due to the lack of a safe and effective vaccination or other treatment option, control relies exclusively on veterinary hygiene measures, i.e., the killing of affected herds, the establishment of restriction zones and intensive surveillance [3]. The disease is subject to compulsory notification.

The viral haemorrhagic disease of suids is caused by a double-stranded DNA virus of the genus *Asfivirus* within the *Asfarviridae* family, African swine fever virus (ASFV). Its genome has a size of approx. 190 kbp and encodes for more than 150 proteins, including factors that modulate the host immune response [4]. About half of the ASFV genes still lack any known or predictable function [5,6].

The virion has a very complex structure and an overall diameter of 175–215 nm. So far, it has been acknowledged that the virion consists of a nucleoprotein core, a core shell surrounded by an internal lipid layer, an icosahedral capsid, and a dispensable lipid envelope [4,7]. However, single-particle cryo-EM analyses of the three-dimensional structure of the ASFV particle have revealed recently that the nucleoid is in fact surrounded by two distinct icosahedral protein capsids and two lipoprotein membranes, one following the icosahedral symmetry surrounding the inner capsid, one the outer, and originating from the budding process [8].

In the vertebrate host, the virus replicates in cells of the mononuclear phagocyte system (in older literature referred to as reticuloendothelial system), predominantly monocytes and macrophages, although several other cell types can be infected, especially in the later stages of the disease [9,10,11,12,13,14].

The clinical picture depends on strain virulence and the immunological status of the affected animal. Upon infection with highly virulent strains, most animals die within 10 days post infection showing high fever, severe but unspecific general signs, haemorrhages in different organs, and pulmonary edema [15]. Thrombocytopenia, lymphopenia, and monocytopenia are observed [16]. It is generally accepted that proinflammatory cytokines are involved in the immunopathology of ASF [17], including the development of fever and changes in levels of acute-phase proteins [16].

Animals surviving infection can recover completely and are protected against reinfection with similar strains, but so far, no stringent correlates of protection could be deduced [18]. The neutralizing capacity of antibodies against ASF remains controversially discussed [19]; nonetheless, it is clear that antibodies alone do not sufficiently protect animals against infection [20], as is also seen with inactivated vaccines that are immunogenic but not protective [21,22]. Thus, cellular immunity plays a major role, and T cell responses are particularly crucial [23].

## 2. Antigen-Presenting Cells: Bridging Innate and Adaptive Immunity

Macrophages (Mφ) and dendritic cells (DCs) are professional antigen-presenting cells (APCs). They are equipped with a vast array of pathogen recognition receptors (PRRs), including Toll-like receptors (TLRs), which detect highly conserved microbial molecular signatures, named pathogen-associated molecular patterns (PAMPs) [24,25,26]. These cells are responsible for bridging innate and adaptive immunity. After detection and uptake of foreign molecules, they process and present antigens to T lymphocytes, driving the development of an antigen-specific acquired immune response [24,25]. For an appropriate T cell activation, three different signals are required: (1) through T cell receptor (TCR) interaction with antigen-presenting MHC molecules termed swine leukocyte antigens (SLAs) in pigs, (2) by costimulatory molecules, and (3) through cytokines [27].

Alongside conventional APCs, there are nonconventional APCs, such as γδ T and natural killer (NK) cells. NK and γδ T cells are two major lymphocyte populations of the innate immune system, which also play a role in the activation and regulation of acquired immune responses [28,29]. Both cell types can display characteristics of professional APCs: they can process and present antigens to naïve T lymphocytes via SLA class II molecules, thus promoting the development of acquired immune responses [29,30,31].

### 2.1. Macrophages

Macrophages are professional phagocytic cells, which detect and clear foreign bodies, and are able to drive the development of acquired immune responses [25,32]. The main target for ASFV are Mφ, which seem to play a key role in the immunopathogenesis of ASF [33]. The virus evolved numerous strategies to inhibit Mφ defences and to recruit several elements of the Mφ translation machinery to viral factories in order to replicate efficiently in these cells [34].

SLA class I and II are involved in antigen processing and presentation [35]; thus, several studies analysed virus-driven modulation of their expression [36,37,38,39]. In fact, changes in SLA class I or II expression could affect antigen-presentation and thus the development of adaptive immune responses. The expression of SLA class II on porcine alveolar macrophages (PAMs), bone marrow-derived macrophages (BMDMs), or monocyte-derived macrophages (moMφ) is not modulated by ASFV [36,38,39], whereas ASFV strains of diverse virulence differently affect SLA class I expression on moMφ. Netherton and colleagues observed that virulent Malawi Lil/20 ASFV increased the expression of SLA class I genes in porcine moMφ but with no parallel increase in SLA class I molecule delivery to the plasma membrane, due to ASFV ability to disrupt vesicular transport at the trans-Golgi network [40]. In addition, differences between ASFV strains of diverse virulence were observed: attenuated genotype I ASFV (BA71V, NH/P68), but not virulent genotype I ASFV (22653/14, 26544/OG10), downregulated SLA I expression on infected moMφ [36,37]. SLA class I downregulation induced by attenuated ASFV might enable NK cell activation in vivo.

In fact, these cells are triggered to kill or ignore potential targets depending on a balance between inhibitory and activating signals, and SLA class I molecules provide a strong inhibitory signal for NK cells [41]. A previous study supported this hypothesis: domestic pigs were inoculated with NH/P68, and then researchers observed a correlation between NK activation and protection to challenge with the homologous virulent L60 [42]. Through the impairment of SLA class I downregulation, virulent genotype I ASFV isolates might elude early NK recognition, with subsequent poor development of a protective acquired immune response.

Various ASFV strains of diverse virulence not only differently modulate SLA class I expression but trigger a different pattern of soluble mediators (cytokines and chemokines) released by Mφ. Attenuated genotype I ASFV strains (NH/P68, OURT88/3) induced enhanced expression of key cytokines (IFNβ, several IFNα subtypes, IL-1β, IL-12p40, TNF-α) and chemokines (CCL4, CXCL8, CXCL10) compared with highly virulent isolates (L60, Benin 97/1, 22653/14, Armenia2007) [37,43,44,45,46,47,48,49].

IFNα and IFNβ are crucial cytokines in the fight against viral infection that inhibit the replication of many RNA and DNA viruses [50]. These molecules also present several immunomodulatory properties, such as promoting DC maturation [51], inducing SLA II upregulation in conventional and unconventional APCs [52,53], increasing NK cytotoxic activities, and promoting Th1 differentiation [51]. Virulent ASFV present several genes within multigene families MGF360 and 530/505 involved in inhibiting type I IFN induction in ASFV-infected Mφ [47,48,54], with a consequent escape from immune surveillance.

IL-1β and TNF-α are proinflammatory cytokines, which are released at the early stages of the immune response [55]; IL-1β also enhances antigen-driven expansion and differentiation of CD4 T cells [56]. IL-12 is a key inducer of IFNγ production and promotes Th1 and NK cell cytotoxicity [55]. CCL4, CXCL10, and CXCL8 are chemokines able to attract mononuclear cells to sites of inflammation, and CCL4 can also promote the development of IFNγ producing Th1 lymphocytes [57]. The null release of proinflammatory or pro-Th1 chemokine/cytokines by Mφ infected with virulent ASFV isolates probably weakens immune vigilance. On the contrary, the release of these soluble mediators by Mφ infected with attenuated ASFV strains might enhance immune surveillance and promote the development of effective adaptive immune responses.

Another study investigated gene expression changes in moMφ after ASFV infection through transcriptome analysis. Researchers observed that moMφ’s infection with virulent Georgia 2007 resulted in downregulated expression of anti-inflammatory IL-10 and upregulated expression of IL-17 and cytokines of TNF superfamily, including FASL, LTA, LTB, TNF, TNFSF4, TNFSF10, TNFSF13B, and TNFSF18 [58]. Authors speculated that in vivo, the release of these cytokines might stimulate the death of bystander cells and tissue inflammation [58]. Others observed that both attenuated NH/P68 and virulent 22653/14 ASFV downregulated IL-10 expression [37], although insufficient data are available on whether ASFV strains of diverse virulence differently modulate induction or release of IL-17, FASL, or other cytokines of the TNF superfamily. Different levels of these cytokines might affect ASFV pathogenesis; thus, this may constitute the object of future studies.

In addition, several studies suggest that ASFV infection affects the ability of these cells to respond to external stimuli [36,37]. Infection with ASFV leads to CD14 and CD16 downregulation on moMφ [36] and BMDMs [38], with potential impairment of Mφ’s antimicrobial and antiviral activities. We recently observed that infection with either attenuated (NH/P68) or virulent (22653/14) ASFV resulted in moMφ impaired ability to release IL-12, IL-6, and TNF-α in response to stimulation with IFNγ and LPS or a TLR2 agonist [37]. Overall, these data suggest that the virus has developed various mechanisms to overcome Mφ defenses in order to use these cells for its efficient replication.

To date, little is known about ASFV interaction with polarised Mφ. Mφ are a heterogeneous population with remarkable plasticity, changing their phenotype and functions in response to the subtle and continuous changes of surrounding signals [59]. The two extremes of diverse polarised status are represented by ‘classically’ (M1) and ‘alternatively’ (M2) activated Mφ, with M1 characterised by antimicrobial and tumoricidal functions, and M2 mainly related to immunosuppression and wound healing processes [59]. M2 polarisation did not affect macrophage responses and susceptibility to ASFV infection, whereas M1 activation of either PAMs or moMφ resulted in delayed ASFV replication [37,60]. A recent study showed that ASFV infection in PAMs resulted in increased Arg-1 expression [61], which is a hallmark of M2 polarisation in many species, including swine [62]. In addition, the authors observed that L-Arginine promoted ASFV replication in PAMs [61]. ASFV probably promotes M2 polarisation to overcome host defences, considering that M1 macrophages present greater antimicrobial functions and promote T cell response development, which in vivo would limit ASFV dissemination in the host. Differences between strains of diverse virulence were also observed. The attenuated NH/P68 presented a reduced ability to infect moM1 compared with moMφ, and moM1 released IL-1α, IL-1β, and IL-18 in response to infection with this attenuated strain. In contrast, moM1 infection with virulent 22653/14 ASFV did not induce the release of either proinflammatory (IL-1α, IL-1β, IL-6, TNF-α) or anti-inflammatory (IL-10) or pro-Th1 (IL-12, IL-18) cytokines [37]. As above stated, IL-1β is a proinflammatory cytokine, and its release might foster apoptosis of uninfected bystander cells, thus reducing ASFV spread into the host. IL-18 is indeed an inducer of IFNγ production, and it activates T cells and NK cells in synergy with IL-12 [55]; thus, its release in response to infection with attenuated strains might promote T cell responses. The potential impact of alterations of Mφ functionality by IL-10 during ASFV infections are also not well understood [63].

These data suggest that virulent ASFV isolates developed strategies to elude macrophage immune surveillance, with a consequent abrogation of a protective acquired immune response. These decoy strategies are partially defective in attenuated strains.

### 2.2. Dendritic Cells

Dendritic cells (DCs) are regarded as sentinels of the immune system, able to process and present antigens to naïve T lymphocytes, and with a central role in activating and shaping adaptive immune responses against pathogens [24]. Porcine DCs are a heterogeneous population with organ-specific peculiarities, recently reviewed in [64]. Porcine DCs can be divided into two main subsets: conventional DCs, involved in priming a Th response, and plasmacytoid DCs (pDCs), able to release high amounts of IFNα in response to viral infections [65].

To overcome the low frequency of DCs in both blood and tissues, cDCs can be conveniently generated in vitro from monocytes, using a medium supplemented with recombinant IL-4 and GM-CSF (monocyte-derived DC, moDC) [66]. A limited number of studies investigated moDC interaction with ASFV. Several ASFV strains of diverse virulence (NH/P68 and 22653/14) efficiently replicated in moDCs, and infection with attenuated NH/P68 ASFV resulted in SLA I downregulation, as observed in moMφ. Instead, infection with the virulent 22653/14 ASFV did not alter any surface marker expression and did not trigger the release of either IFNβ, anti-inflammatory IL-10, proinflammatory (IL-1β, IL-6, IL-8, TNF-α), and pro-Th1 (IL-12, IL-18) cytokines [67]. Another study reported that moDCs infection with attenuated OURT 88/3 or virulent OURT 88/1 ASFV isolates led to the reduced phagocytic ability of these cells [68]. These preliminary data indicated that ASFV developed mechanisms to escape cDC defences [64]; however, to date, no study investigated ASFV interaction with bona fide cDCs. Only one study evaluated pDC responses to ASFV: PBMCs enriched for DCs released great amounts of type I IFN after ASFV infection [49]. Researchers speculated that pDCs were likely the source of the abnormally high levels of type I IFN observed in pigs during acute ASFV infection [49]. Excessive type I responses are correlated with lymphopenia in many viral infections [50], which results in an inability to mount an adequate protective immune response.

To date, only a few in vivo studies on the role of DCs during ASFV infection have been carried out. Viral antigens were detected in interdigitating DCs (iDCs) in mandibular lymph nodes of domestic pigs three days postinfection with the virulent ASFV L60, and this was succeeded by a reduction in the number of iDCs. The authors speculated that an early DC depletion in lymph nodes of pigs infected with virulent ASFV might impair the development of a protective, adaptive immune response [69]. More recently, it was observed that ASFV late viral proteins (p72) were detected in DCs in several tested organs (spleen, lungs, liver, and lymph nodes) of pigs and wild boar infected with the moderately virulent Estonia 2014 [70]. Future research should address whether ASFV can directly infect DCs or whether these cells acquire exogenous antigens from other infected cells. In fact, DCs are also able to process exogenous antigens through the SLA class I pathway by a process named ‘cross-presentation’ [71]. This process can be extremely important in viral infections where the pathogens evolved strategies to impair DCs immune functions [71].

### 2.3. Nonconventional APCs

Both porcine γδ T cells and NK cells can detect and attack pathogen-infected cells and are able to produce IFNγ and other immunostimulatory cytokines [29]. Their contribution to protection as effector cells will be better discussed later in this review (see Section 3.2 and Section 3.3). Nevertheless, as above stated, they can also act as professional APCs [29,30,31,72]. SLA II is regarded as an activation marker for invariant Natural Killer T cells (iNKT), and an increase in this cell subsets was observed after ASFV infection [73]; however, NK cells function as APCs during ASFV infection were never investigated. On the contrary, it was observed that γδ T cells were able to present ASFV viral antigen to T cells [74]. Considering that the virus developed several strategies to inhibit both macrophages and DCs responses, the role of other innate immune cells in antigen presentation should be thoroughly investigated in the future. 

## 3. T Cell Responses against ASFV Infection

In contrast to the innate immune system, which elicits fast but unspecific responses, the adaptive immune system is capable of mounting precise responses against viral infections and establishing immune memory in their aftermath that protects in case of recurring infections. This immunity against future infections can be conferred by humoral and cellular responses, i.e., B cell and T cell responses, respectively. However, although ASFV vaccine candidates often induced neutralising antibodies (nAbs), they alone were not sufficient to confer protection from lethal challenge infections. DNA vaccination with ASFV-derived hemagglutinin (sHA, also CD2v), p54, and p30 was not protective. However, when the construct was fused to ubiquitin to increase the presentation by SLA I, it induced protective T cell responses even in the absence of nAbs [75] (for more details, see Section 4.3). Similarly, lack of protection against heterologous challenge infections induced by a live attenuated vaccine (LAV) candidate correlated with an absent response of CD8α+ lymphocytes [17]. However, there are also studies indicating a protective role for humoral responses against ASFV infection, shown by the survival of lethal challenge infections after the transfer of convalescent plasma or colostrum from convalescent sows [76,77]. These responses are still not entirely understood and will not be further discussed in this review but have been reviewed in the past [19]. Instead, these and several other studies over the years demonstrated the essential role of T cell responses for the clearance of ASFV infections. A vast body of literature exists about T cell responses in humans and mice, but porcine T cell responses remain understudied, especially those against nonzoonotic threats such as the ASF. In this section, we will summarise the currently existing knowledge and highlight gaps in our understanding of porcine T cell responses against ASFV infections and during immunisations.

### 3.1. αβ T Cells

Hallmarks of all mammalian T cells are the expression of CD3 molecules and a TCR, which together facilitate antigen detection and cell activation. The TCR consists of either α and β chains or γ and δ chains in αβ and γδ T cells, respectively. Depending on the age and environmental factors, the major subset of porcine CD3+ T cells is αβ T cells. Porcine αβ T cells are subdivided into three subsets: CD4+ T helper (Th) cells, CD8α+ cytotoxic T cells (CTLs), and, in contrast to humans and mice, a significant population of CD4+/CD8α+ double positive (DP) T cells, usually thought of as effector and memory cells [78].

This section will discuss the properties and contributions of these subsets to immunity against ASFV infections.

#### 3.1.1. CD8α+ αβ T Cells and Cytotoxic Responses

CD4–/CD8α+ αβ T cells are typically considered CTLs. They recognise their respective antigen in the context of SLA class I and, thus, respond to intracellular antigens. By killing infected cells, CTLs inhibit the spread of obligate intracellular pathogens such as viruses.

Initial evidence for an involvement of (cytotoxic) T cells was found in studies in the early 1980s, where peripheral leukocytes from pigs infected with an attenuated ASFV strain displayed specific cytotoxicity against ASFV-infected cells [79,80]. Moreover, proliferation and even persisting cytotoxic responses after in vitro restimulation indicated the successful formation of T cell memory in animals infected with attenuated ASFV strains up to two years after the initial infection [81], while animals infected with virulent ASFV strains died before memory cells could have formed [80]. Extensive in vitro studies by Martins et al. further characterised the CTL responses in animals surviving low virulent ASFV infection [82]. The researchers infected pigs with the low virulent, nonhaemadsorbing NH/P68 (NHV, [83]). PBMCs from these ASFV-immune pigs were isolated and restimulated with low doses of NHV and then investigated in CTL assays with NHV-infected porcine macrophages as target cells. These assays showed a significant specific lysis and, thus, CTLs activity. Further studies with SLA-mismatched PBMCs and blocking mAb against SLA I (clone PT85a) showed reduced specific lysis. Finally, when PBMCs were depleted of either CD4+ (mAb clone 74-12-4) or CD8α+ (mAb clone 76-2-11) cells, specific lysis was significantly reduced in CD8α-depleted samples only [82]. These ex vivo studies marked the first descriptions of protective immunity conferred by CD8α+ CTLs.

Oura and colleagues undertook a study that demonstrated the importance of CD8α+ T cells most evidently so far [23]. After infection with and clearance of the low virulent OURT88/3, surviving animals were depleted of CD8α+ lymphocytes with specific antibodies (clones 76-2-11 and 11/295/33). Subsequently, the animals were challenged with the homologous but highly virulent OURT88/1. Depleted animals succumbed to the challenge infection, while nondepleted animals survived and even showed only mild clinical signs of disease [23]. However, since various porcine lymphocyte subsets express CD8α, from classical CD8α+ αβ T cells, activated CD4+ αβ T cells, and γδ T cells [84], to nonconventional T cells [73] and NK cells [72], it remained unknown which cells conferred immunity. Given the previous evidence of the involvement of cytotoxic T cell responses, the researchers tried to specifically deplete CTLs with another mAb (clone PPT22, recognizing the CD8 β-chain) in a second trial otherwise similar to the first. At least partially sufficient depletion was achieved only in one out of ten animals. Coincidently, this animal was the only one succumbing to the challenge infection [23]. Of note, the study design only allowed for the investigation of memory responses, while primary responses in the acute phase of ASF have not been studied. Still, both trials in this study presented the first definite in vivo evidence of the pivotal contributions of CD8α+ T cells to immune memory against ASFV.

More recently, cytotoxic responses were investigated in comparative studies with domestic pigs and wild boar during two different ASFV infections up to 10 days postinfection (dpi). Infection with moderately virulent Estonia2014 induced significant increases in the frequency of CD8α+ CTLs 5–10 dpi, especially in spleens, lungs, and livers of infected domestic pigs. Wild boar responded similarly and even displayed a T-bet-dependent differentiation 10 dpi, indicating a beginning antiviral Th1 response [85]. During infection with highly virulent Armenia2007, the response was dominated by DP T cells in both subspecies with no changes of CD8α+ CTLs until the end of the study seven dpi [86].

The most unexpected, yet in multiple independent trials reproducible finding, was a nearly abrogated expression of one of the main lytic effector proteins, perforin, in multiple cytotoxic effector populations (CD8αα+ and CD8αβ+ αβ T cells, and CD8α+ γδ T cells) 4-5 dpi during infections with both moderately and highly virulent ASFV strains. This effect was most evident in domestic pigs during infection with Armenia2007 and less pronounced after infection with Estonia2014. Wild boar showed significant loss of perforin expression as well, but generally to a lesser extent than domestic pigs [85,86].

The reason for this observation remained unknown. Besides perforin, CTLs clear infected cells with other cytotoxic molecules, such as Fas ligand (FasL or CD95L/CD178) or TNF-related apoptosis-inducing ligand (TRAIL), by death receptor-mediated apoptosis instead of lysing cells directly [87]. However, the expression of these molecules cannot be detected by mAbs in pigs yet. It is possible that CTLs in ASFV-infected animals switch to apoptosis induction by FasL or TRAIL instead of lytic cell clearance, which has been shown to result in a total loss of perforin expression [88]. This might prove to be beneficial because some ASFV strains express a viral homologue (A179L) of the mammalian antiapoptotic protein Bcl-2 [89,90]. Antiapoptotic effects of Bcl-2 are most effective against perforin-mediated killing [91]. Lower perforin levels in domestic pigs might indicate a switch to death receptor-mediated killing, while a more perforin-mediated response in wild boar could have been counteracted by the virus. Antiviral perforin responses in the liver have also been shown to cause substantial tissue damage [92], giving a possible explanation for the observed Kupffer cell degeneration in wild boar and overall differences in disease outcome [70]. Another explanation is perforin consumption, either because of a hyperinflammatory response or T cell exhaustion. Immediate secretion of synthesised perforin, which cannot be detected by the mAb (clone δG9) used in the studies [93], indicates a proinflammatory, polyclonal activation. Strong inflammatory signals and high antigen loads during acute infections lead to continuous T cell activation that results in T cell exhaustion [94]. However, the exact mechanisms remain to be elucidated.

#### 3.1.2. CD4+ and CD4+/CD8α+ αβ T Cells

CD4+/CD8α– αβ Th cells are aiding in affinity maturation, antibody production, and induction and orchestration of further immune responses. Porcine CD4+ T cells that become activated or differentiate into memory cells express CD8α [78], a characteristic rarely seen in extrathymic human or murine T cells of healthy individuals but not unique to pigs, as it has been shown in other species such as dogs, chicken, and monkeys [95,96]. CD4+/CD8α+ DP T cells will, therefore, be discussed in this chapter as well.

Given the abovementioned initial evidence for the importance of CTL responses, CD4+ Th cells have rarely been investigated in detail. However, some studies also provided evidence for an evolving CD4+ Th cell response with important duties during ASFV infections. In an early study, PBMCs from pigs that survived infection with the attenuated E75 ASFV strain were incubated with live or UV-inactivated ASFV, i.e., intra or extracellular antigens. While the live virus-induced proliferation of both CD4+ and CD8α+ T cells, UV-inactivated ASFV stimulated CD4+ T cells only. Similarly, blocking mAbs against CD4 (clone 74-12-4) or SLA II (clone MSA3) completely abolished proliferation in samples stimulated with UV-inactivated ASFV, while mAbs against CD8α (clone 76-2-11) and SLA I (clone 74-11-10) only partially decreased proliferation [97]. These results indicated an involvement of CD4+ Th cells, although differentiation between CD4+/CD8α– and DP T cells was not possible at the time. Functional involvement of CD4+ Th cell responses was shown in a study with PBMCs from pigs immunised with the attenuated BA71 ASFV strain [98]. Following in vitro restimulation with ASFV, antibody secretion from these PBMCs was investigated. Interestingly, antibody production was nearly abrogated when all T cells were removed by rosetting or when CD4+ Th cells were selectively depleted by complement-dependent cytotoxicity with anti-CD4 mAbs (clone 74-12-4) [98]. These findings point to a dependency on CD4+ Th cells for the efficient production of antibodies during ASFV infection and maybe even a T cell-dependent class switch.

Of note, the aforementioned depletion experiments by Oura et al. likely depleted DP T cells that differentiated during the primary ASFV infection as well [23]. A clear distinction between the consequences of depleting either CD4−/CD8α+ or DP T cells is consequently impossible. However, together with other studies showing the development of DP T cells after ASFV infection, it emphasises a role for DP T cells in secondary responses against ASFV.

Given the fact that DP T cells develop over days and weeks throughout the course of an infection, studies investigating the role of DP T cells for immunity usually focused on memory responses. However, the aforementioned comparative studies with highly virulent Armenia2007 and moderately virulent Estonia2014 ASFV infections in domestic pigs and wild boar also allowed for investigation of primary T cell responses [85,86]. During Armenia2007-infection, domestic pigs showed a significant increase in DP T cells 5–7 dpi in all investigated tissues (blood, spleen, gastrohepatic lymph node, liver). Interestingly, these cells were not proliferating, as shown by the absence of Ki-67 expression. Together with the inverse correlation of observed CD4+ T cell frequencies, this indicated that the DP T cells found in infected animals differentiated from CD4+/CD8α– Th cells. DP T cells in wild boar, on the other hand, proliferated massively but showed no significant increases during the study. These cells might have migrated to tissues not investigated in this study or have undergone apoptosis during antiviral responses [86].

Moderately virulent Estonia2018 induced increasing frequencies of DP T cells in spleens, lungs, and livers of domestic pigs and in lungs and livers of wild boar. The proliferation of DP T cells was only found in spleens of infected domestic pigs [85]. Interestingly, the loss of perforin that was observed in CTLs was not found in DP T cells. Since proliferating DP T cells were only found in the spleen, the authors speculated that DP T cells exert orchestrating but not cytotoxic functions during moderately virulent ASFV infection [85]. This interpretation is in contrast with earlier findings of perforin upregulation in DP T cells after infection with the low virulent OURT88/3 [99]. However, besides different strains used for infection, the latter study was also conducted in only two pigs.

Together, these studies point to a significant role of DP T cells not only as memory cells protecting against recurring infections but also as evolving effector cells during primary ASFV infections.

#### 3.1.3. Regulatory T Cell Responses

Regulatory T cells (Tregs), defined in pigs as in other mammalian species as CD3+/CD4+/CD25+/FoxP3+ [29], are a subset of CD4+ αβ T cells but will be discussed in this separate chapter because of their distinct role in immunity. A hallmark of Treg responses is the secretion of the immunomodulatory cytokine IL-10, which inhibits proinflammatory Th1 cytokines and antigen presentation but at the same time augments B cell responses and antibody production [100]. Responses of Tregs in ASFV-infected pigs are not entirely understood, mostly because infected animals succumb to infection or have to be euthanised before Treg responses evolve. However, there are still several studies with ASFV and other viruses indicating a central role of Tregs in the modulation of antiviral responses in pigs. Porcine Tregs have been shown to facilitate viral persistence during infection with porcine reproductive and respiratory syndrome virus (PRRSV) by secretion of IL-10 and subsequent inhibition of Th1-mediated antiviral responses [101]. In line with this, several ASFV vaccination trials by Sánchez-Cordón et al. found increasing levels of IL-10 in nonresponding pigs that succumbed to challenge infection [102,103,104]. Some of these studies also found increased levels of IFNγ in the same pigs, which might indicate a generally derailed immune response in moribund animals [102,103].

Several infection studies described a negative correlation between secretion of IL-10 and survival. Cabezón et al. compared a group of healthy wild boar with another group infected with Classical swine fever virus (CSFV) shortly after birth. Both groups were infected with the moderately virulent E75 ASFV strain at seven weeks of age. All CSFV-positive animals died within six or seven dpi. Interestingly, only the CSFV-negative wild boar secreted detectable levels of IL-10, and two of these animals died eight dpi, while one animal survived the study period of 10 days [105]. In another study, Barroso-Arévalo et al. showed that wild boar protected by vaccination exhibited no increase in secreted serum IL-10 after contact to infected, shedding animals, while unvaccinated controls were found to have significantly increased IL-10 levels and later died of ASFV infection [106]. Similarly, Reis et al. found increased levels of IL-10 only in animals that died from challenge infection with the virulent OURT88/1 ASF strain after immunisation with an I329L deletion mutant based on the attenuated OURT88/3 strain (OURT88/3ΔI329L) [107]. Other studies from the same group found no significant changes in IL-10 levels after immunisation or challenge infections with different ASFV strains [47,108]. A small study by Wang et al. also found IL-10 only in animals on the day of or shortly before death from ASFV infection [109]. Taken together, these studies suggest that the occurrence of IL-10 is not part of a physiologically orchestrated immune response but rather a sign of a fatally derailed system that will not recover. Consistent with these findings, recent studies of domestic pigs and wild boar found Tregs earlier and at higher frequencies in wild boar after both moderately and highly virulent ASFV infection, correlating with increased disease severity and lethality [73,86]. However, no cytokines were measured in these trials.

In contrast, a report by Post et al. correlated increasing IL-10 levels with the survival of moderately virulent ASV infections [110]. However, given the relatively low levels of IL-10, as well as low numbers and higher age of survivors compared with animals that succumbed to ASFV infection in this analysis, the results seem less robust and might instead be attributed to age differences.

### 3.2. γδ T Cells

Contrary to activation of αβ T cells by conventional TCR-MHC interaction, the γδ TCR, although able to bind to MHC molecules [28], is thought to work similarly to PRRs [111,112]. Moreover, even viral proteins on infected cells have been identified as ligands for γδ TCRs [113,114]. Pigs, as other artiodactyls and in contrast to humans or mice, have high numbers of circulating γδ T cells [115]. Three porcine γδ T cell subpopulations have been described based on the expression of CD2 and CD8α: naïve (CD2–/CD8α–), activated (CD2+/CD8α–), and terminally differentiated effector (CD2+/CD8α+) γδ T cells [116,117]. Given their functional plasticity and high numbers in pigs, some studies also investigated the role of γδ T cells during ASFV infection.

In a study with domestic pigs of different ages (12 and 18 weeks), heightened frequencies of γδ T cells correlated with increased survival after infection with a low or high dose of a moderately virulent Netherlands’86 [110]. Earlier studies showed that porcine γδ T cells are generally capable of presenting antigens to other T cells [118] and that they restore ASFV-specific proliferation in the absence of professional APCs [74]. Therefore, it might be speculated that the assumed protection is attributable to increased antigen presentation by γδ T cells, which compensates for the impaired antigen presentation by professional APCs in the early phase of ASFV infection. However, the survivor groups were small, and the effect was most prominent when γδ T cell frequencies were correlated with age. Twelve-week-old pigs also died earlier independently of the inoculation dose, indicating that age and immune maturity, rather than specific cell frequencies, were the predominant factor for survival [110].

The hypothesis of a protective role for γδ T cell responses is further challenged by studies of ASFV infections in wild boar, which consistently exhibited a T cell response dominated by γδ T cells but also more severe clinical signs and a higher lethality after experimental infection [70,85,86,119]. Infection with the highly virulent ASFV strain Armenia2007 resulted in increased γδ T cell frequencies or differentiation into CD8α+ effector cells in the blood, gastrohepatic lymph nodes, and spleens of infected wild boar, while no such response was detectable in domestic pigs [86].

Differences between domestic pigs and wild boar became even more obvious in trials with the moderately virulent Estonia2018. While there was no substantial γδ T cell response in domestic pigs throughout the study, wild boar demonstrated a significant bias towards γδ T cell responses and showed increased frequencies of differentiated CD8α+ γδ T cells 4–7 dpi [85]. Interestingly, this response was most evident in livers of infected wild boar, correlating with histopathological findings of severe Kupffer cell degeneration 7–10 dpi in infected wild boar [70]. A significant increase in T-bet+ T cells was also only found in infected wild boar [85]. Contrary to previous studies, these findings indicate a potentially pathological immune response driven by γδ T cells and might explain the increased lethality of wild boar in an experimental infection with Estonia2018 [70,119].

However, the different outcomes of ASFV infection in domestic pigs and wild boar might also be explained by inherent differences in the immune responses of both suid subspecies, which are currently virtually unexplored.

### 3.3. Nonconventional T Cells

There are other populations of T cells that have distinct features discriminating them from conventional αβ and γδ T cell populations. A well-known and extensively studied population are invariant Natural Killer T (iNKT) cells. These cells have been shown to have central characteristics found in all T cells: development in the thymus, expression of CD3 and a TCR, and generation of antigen-specific responses after activation. However, they also have a few special features that make them distinct from conventional T cells. Besides an antigen-experienced phenotype already after egress from the thymus, iNKT cells possess an eponymous invariant TCR with decreased variability, detecting a strikingly reduced variety of antigens, usually glyco- and phospholipids, or α-galactosylceramide (αGC) in research settings, in the context of the MHC-related CD1d. Moreover, iNKT cell responses can also be induced by cytokines such as IL-12, IL-18, and type I IFN [120]. A number of studies have investigated porcine iNKT cells [73,121,122], but data about their role in ASFV infections are scarce.

Responses of NKT cells during ASFV infections have been assumed in early studies, where lymphocytes comparable to NKT cells (CD3+/CD4–/CD5±/CD6–/CD8α+/CD11b+/CD16+/perforin+) expanded after in vitro culture of naïve PBMCs with SLA-matched ASFV-infected cells [99,123]. Although a part of these cells was indeed αGC-reactive [123], their frequency was too high to consist of iNKT cells only. By using a murine CD1d tetramer loaded with the αGC analogue PBS57 for the direct identification of porcine iNKT cells [121,122], later studies showed a response of iNKT cells during ASFV infection in vivo. The iNKT cell frequency was significantly increased five dpi in the liver, gastrohepatic lymph nodes, lungs, and peripheral blood of domestic pigs infected with the highly virulent Armenia2007 [73]. Interestingly, although a significant in vivo response was found, there was no activation when porcine PBMCs were incubated with Armenia2007 in vitro. This has been attributed to immune escape mechanisms elicited by virulent ASFV strains, such as blocked expression of type I IFN [44], which could be counteracted in vivo to some extent. Moreover, in a study with the moderately virulent Estonia2014, an increase in ICOS+ iNKT cells 7–10 dpi in blood and spleen has been found [85]. Expression of ICOS has previously been attributed to a high activation status of iNKT cells in mice and pigs [73,124,125]. Together, these studies indicate that porcine iNKT cells take part in antiviral responses in domestic pigs. Given that iNKT cells are potent producers of IFNγ [120], a protective role can be hypothesised. However, the total extent and impact of iNKT cell responses require more research.

Besides iNKT cells, another nonconventional T cell population are mucosal-associated invariant T (MAIT) cells. These have been found in several species and were extensively investigated in mice and humans. Similar to iNKT cells, they were implicated to play a role in antiviral responses against numerous viruses [126]. In pigs, however, there has only been one report of cells expressing transcripts of the invariant MAIT cell TCR chain, TRAV1-TRAJ33 [127]. Cellular identification and analyses of their responses against infectious diseases in pigs are still missing.

## 4. Recent Developments and T Cell Epitopes in ASFV Vaccines

### 4.1. Protection and Induction of IFNγ by Recent ASFV Vaccine Candidates

The pivotal role of cellular adaptive responses and the repeatedly demonstrated lack of protection by nAbs against ASFV alone [20,23] emphasised the need to analyse T cell responses throughout vaccine trials. However, because a conclusive correlate of protection is still missing, most researchers have investigated the protective potential by detecting surrogate markers, e.g., secretion of IFNγ. This is, in itself, still debatable because numbers of IFNγ-producing cells or IFNγ levels did not correlate with survival of challenge infections in several studies [18,47,128,129]. Since potential vaccines against ASFV have been extensively reviewed by Muñoz-Pérez and colleagues recently [130], we will focus on candidates that elicited a T cell response in this chapter.

Vaccines using live-attenuated viruses generally showed superior effectiveness in comparison with compositions with inactivated viruses [22], presumably because the latter fail to induce protective cytotoxic responses due to a lack of SLA I presentation [97]. The live-attenuated ASFV-G-ΔMGF, lacking six genes of the multigene family MGF360 and MGF505, protected pigs against challenge with the virulent parental ASFV-G [131]. Similarly, the attenuated BeninΔMGF also showed induction of protective immunity in pigs against homologous challenge with a lethal dose of Benin97/1, albeit with prolonged viremia. Interestingly, when compared with OURT88/3 that carries identical but fewer deletions of MGF genes, antiviral T cell responses in pigs immunised with BeninΔMGF were reduced. This was most evidently demonstrated by lower numbers of circulating CD8α+ γδ T cells and fewer IFNγ-secreting cells, indicating that protection may be executed by differently induced T cell responses [47]. A virus based on the virulent Pretoriuskop/96/4 lacking the virulence-associated gene 9GL (Pret4∆9GL) was used to immunise pigs before challenge with the virulent parental virus. Protection against challenge infection increased progressively after immunisation, but there was no association between the amount of IFNγ-secreting cells and protection [18].

In vaccine trials using vector viruses coding for ASFV proteins that were shown to induce T cell responses in splenocytes from recovered animals in vitro, there was no consistent correlation between the number of IFNγ-secreting cells and protection against virulent OURT88/1 challenge infection [128,132]. However, these studies identified several immunogenic ASFV proteins (discussed in more detail in the next chapter) and demonstrated the effectiveness of viral vectors as a promising vaccine platform.

### 4.2. CTL Targets

The targets, and the epitopes within those targets, of the ASFV specific T cell response, are still largely unknown. Virulent viruses encode at least 161 open reading frames, and this may be higher as novel short ORFs have recently been identified in an attenuated strain, which may also be present in their virulent counterparts [133]. Cells from pigs that were immunised with low virulent strains of ASFV and subsequently challenged with related virulent viruses have been used to characterise the T cell response to ASFV as well as to begin the identification of potential targets. PBMCs from animals immunised with attenuated E75a or NH/P68 that were depleted of CD4 cells were capable of specifically lysing ASFV infected Mφ. PBMCs from these animals depleted of CD8α+ or SLA-I+ cells did not have this capability [82,134]. CD4 depleted cells from animals immunised with E75a were also capable of specifically lysing cells infected with a recombinant vaccinia virus expressing the CP204L gene that encodes for the highly immunogenic early protein p30 [134]. The results strongly suggest that specific CD8 epitopes are present in CP204L/p30; however, it is important to note that complement-mediated lysis of CD4 cells used in these experiments was not completely effective and that prolonged culture of CD4-depleted cells with ASF virus led to the expansion of a CD8α+ CD4+ population [134]. Later work suggests that a subpopulation of these double positive cells express perforin and have the capacity to specifically lyse ASFV infected cells [123]. In another experiment, PBMCs isolated from pigs infected with the low virulent NH/P68 isolate were capable of specifically lysing macrophages loaded with a 25 amino acid peptide derived from the major capsid protein p72 that is encoded by the B646L gene. B646L/p72 specific lysis could be blocked with antibodies against SLA-I but not SLA-II, again suggesting a role for CD8α+ cells [135]. An additional CTL antigen was identified within the G1340L gene that encodes for a protein with similarity to the vaccinia virus early transcription factor [136]. More recently, PBMCs collected from pigs immunised with pools of adenoviruses expressing the CP204L/p30, B646L/p72, E183L/p54, and CP530R/pp62 genes were shown to induce lymphocytes capable of lysing allogenic fibroblasts transiently expressing the individual open reading frames [137]. Although no specific lysis of ASFV-infected cells was reported in this study, the results suggest that adenovirus-induced CTLs could lyse cells expressing p30 and p72, as was seen in lymphocytes derived from pigs recovered from ASF.

### 4.3. Recall Targets

Response to recall antigen by proliferation or expression of cytokines is commonly used to assess the specificity of the response, and early work showed that stimulation of PBMCs from recovered pigs with the homologous virus can stimulate secretion of IFNγ and interleukin 2, as well as stimulate proliferation [97,134,138]. Blocking experiments with mAbs against CD4 or CD8α showed that both CD4+ and CD8α+ cells proliferated in response to live virus and CD4+ cells also proliferated in response to the inactivated virus [97], and both CD4+CD8α+ and CD4−CD8α+ cells secrete IFNγ after recall stimulation with the live virus [128,139].

A random library containing sequences from the genome of the genotype VIII Malawi Lil 20/1 was used to identify T cell antigens recognised by PBMCs from an animal immunised and challenged with genotype I ASFV. Syngenic fibroblasts were infected with vaccinia viruses expressing the library, and in analogous experiments, proliferation in response to the virus could be blocked by CD8α and SLA-I antibodies, suggesting the assay was biased towards class I presentation. Surprisingly most of the hits identified using this approach were on the opposite strand or in a different reading frame to the annotated gene within that section of the genome [140]. This could be consistent with the presentation of defective ribosomal products hypothesis of class I presentation [141]; however, detailed ASFV transcription and translation maps in macrophages are not available, and therefore it is possible that the sections of the genome that induce proliferation of T cells are from hitherto undefined genes. One of the clones corresponded to a portion of the I329L gene, which encodes for a nonessential immune modulatory gene. As Jensen et al. screened a genotype VIII library against cells from a genotype I immunised animal, this represented the first heterologous antigen identified. However, genotype I immunised animals are not protected against challenge with Malawi Lil 20/1 [142], and the genotype I protein sequence of I329L is 93% identical to that of the genotype VIII equivalent.

A more recent study used a peptide library corresponding to approximately 70% of the proteome of the low virulent OURT88/3 to identify antigens capable of inducing secretion of IFNγ from cells from recovered inbred pigs [128]. A total of thirty-eight ASFV proteins induced secretion of IFNγ from at least one of three inbred pig lines (NIH cc, NIH dd, and Babraham large white). Both p72/B646L and p30/CP204L were recognised by at least two of the different inbred lines, and CP530R, CP312R, and I73R were recognised by all three. A combination of 12 different ASFV genes vectored by replication-deficient adenoviruses were capable of inducing a T cell response that recognised the whole virus in both NIH dd and outbred pigs. A detailed analysis designed to identify the epitopes within these proteins or to phenotype the T cells that recognised them was not undertaken, although some preliminary data suggested that p72/B646L, p30/CP204L, and I73R were recognised by CD8α+/CD4+ cells from animals recovered from low virulent ASFV. It will be interesting to analyse ASFV-specific T cell responses induced by replication-deficient adenoviruses and see if they are comparable to those induced by attenuated strains of the virus.

Antigenic regions and individual epitopes have been identified in a number of ASFV proteins through a combination of fine mapping or bioinformatic-guided discovery with peptides or by eluting peptides directly from SLAs on the surface of infected cells [75,139,143,144]. Peptide mapping identified two regions within the EP153R protein and four within the EP402R/CD2v protein using IFNγ ELIspot [144]. Two of the regions in the EP402R/CD2v protein overlapped with two peptides identified in pigs that had been immunised with a DNA vaccine that included the sequence of the EP402R/CD2v gene [75]. The peptides induced by DNA vaccination were not recognised by the cells from animals recovered from low virulent ASFV; however, it is possible that this was due to differences in the SLA-I alleles encoded by the pigs as the animals were not typed. The two viruses used in these studies, Congo K-49 [144] and E75a [75], are both p72 genotype I but fall into two distinct serogroups [145] and the antigenic region that the peptides from each virus originated from are also the same regions that define the serogroups. This data suggests that the T cell response to EP402R and EP153R may also contribute to the protection mediated by the only antigenic determinate of protection identified within the ASFV genome to date [144,146]. Prediction of SLA-I epitopes within the antigenic sites in the two proteins was not successful, and a large-scale attempt to identify SLA-I epitopes in the Georgia 2007/1 proteome using an exhaustive bioinformatics approach identified a single region in MGF100-1L [139]. Elution of peptides directly from ASFV infected macrophages proved more fruitful, and 56 different ASFV proteins were identified as potential T cell targets with a high degree of conservation between the p72 genotype I sequences identified and the equivalent sequences from genotype II virus. There was no cross-over between the sequences identified using in silico analysis and those determined experimentally. The authors also showed that cells exogenously expressing the ASFV genes MGF505-7R, A238L, or MGF100-1L, stimulated IFNγ from cells from recovered animals [139]. A follow-up study showed that DNA vaccination with plasmids expressing MGF505-7R and M448R could induce T cells that secreted IFNγ after stimulation with live Georgia 2007/1 virus and that this T cell response could enhance the protection offered by a suboptimal dose of a modified live virus vaccine against Georgia 2007/1. This suggested that MGF505-7R and M448R may encode for proteins that are targets of a protective CD8 response. The failure of in silico approaches to identify T cell antigens coupled with the complexity of the ASFV genome means identifying protective and cross-protective epitopes will require a considerable amount of effort. As Bosch-Camós et al. pointed out, Georgia 2007/1 contains at least 50,000 potential 9mers within its 166 predicted open reading frames [139] and the likely presence of minor open reading frames [133,139] and cryptic translation [140] only increases the complexity of the problem.

## 5. Concluding Remarks and Open Questions

Decades of research gave insights into antiviral T cell responses, protective characteristics, and critical aspects for new vaccine candidates (Figure 1).

However, several gaps exist in our knowledge of porcine T cell responses against ASFV infections. Infection of ASFV’s native hosts, warthogs (*Phacochoerus africanus*) and bushpigs (*Potamochoerus larvatus*) [147], typically do not cause clinical signs of infection [148]. Despite knowing of this relationship for long, our understanding of antiviral responses in these species is limited. This is mostly due to the lack of samples, suitable infrastructure, as well as the difficulties of handling wild animals, which makes immunological research in wild suid species difficult. Thus, it still remains unknown whether the reduced clinical burden in warthogs and bushpigs is the result of an efficient viral clearance or (immune) tolerance to disease.

There is also limited understanding of the induction of T cell responses by cells of the innate immune system, such as polarised Mφ. Various agonists, such as IFNβ, or immunosuppressive cytokines (IL-10, TGF-β) can alter porcine Mφ phenotype and functionality [62,63], and future studies should address how different macrophage subsets respond to ASFV strains of diverse virulence. In addition, researchers should investigate whether ASFV infection of DCs leads to impairment of these cells’ activity or whether DCs are still able to prime naïve T lymphocytes with subsequent development of acquired immune responses. More studies focused on the interaction of ASFV with different subsets of this heterogeneous family, as cDC1, cDC2, pDC, or DCs from different tissue compartments, would be valuable to better understand ASFV immunopathology.

In Europe and Asia, wild boar are considered pivotal for the spread of ASFV because of their increasing populations and the absence of ASFV-transmitting ticks in most parts of these regions [70,149,150,151,152]. Still, little is known about natural infections as well as several other factors, e.g., population density or ASFV persistence in the populations, related to the spread of ASFV in wild boar [153]. Antibodies specific for ASFV found in hunted wild boar in Eastern European countries and Sardinia, suggested that at least some animals survive field infections [154,155,156,157]. However, experimental infections of wild boar are often fatal and do not mirror this situation well [119]. Moreover, very little is known about the immune system of wild boar in general. A high genetic consensus between domestic pigs and wild boar suggests similar responses [158]. So far, immunological studies with wild boar focused either on infection with *Mycobacterium* spec. [159,160,161] or classical swine fever virus [105], or on differences of immune-related genes between domestic pigs and wild boar [161,162]. Recently, the first pioneering comparative studies investigated domestic pigs and wild boar infected with ASFV [70,85,86]. Interestingly, although clinical development, viral loads, and pathological findings were largely comparable [70], the mounted immune responses differed: while responses in domestic pigs were dominated by CD8α+ CTL and DP αβ T cells, wild boar showed a distinct bias towards γδ T cell responses [85,86]. However, it remains unknown whether these different responses are based on inherent differences, as even basic data about leukocytes in wild boar is missing.

The mechanisms behind cytotoxic responses against ASFV infections in pigs should also be investigated in greater detail. Which T cell populations exercise protective responses, and which effector molecules exert the highest antiviral effect? Is there a switch from perforin-mediated cytotoxicity to other effector pathways, or is the observed loss of perforin expression based on consumption and, thus, a potential cause of immunopathology? Basic data about the induction of cytotoxic (memory) responses might also lead to more targeted vaccination approaches.

The role of unconventional T cells, especially γδ T cells, should be specifically addressed in future studies. These cells are versatile effectors, and the discussed studies indicate a significant role in ASFV immunity. Special attention should be put on their potential role for the rescue of impaired antigen presentation in myeloid cells and induction of further responses, as indicated in earlier pilot studies [73,74,99,118]. This applies to iNKT cells as well, as increasing frequencies of activated iNKT cells have been found during studies of ASFV infections in domestic pigs [73,85].

The impact of regulatory responses during primary and memory responses is also not well understood, and the available studies do not allow general conclusions. Moreover, future studies of ASFV-infected animals should focus on defining the source of IL-10, which can be secreted by a variety of innate and adaptive immune cells [100].

A distinct correlate of protection is still missing. This would serve not only as a tool to monitor vaccine effectiveness during vaccination trials but could also reduce animal trials and the need for challenge infections with a high disease burden. However, the discussed studies demonstrated that IFNγ secretion is not a useful correlate of protection, as animals with high IFNγ levels often still succumb to disease [18,47,104,128,129,163] and could also be the result of a derailed immune response in moribund animals [102,103]. Moreover, the secretion of IFNγ alone might be misleading, as several studies found IFNγ secretion ex vivo in immunised animals, which failed to translate into in vivo protection after challenge [128,164].

In contrast, some studies did find correlations of IFNγ secretion and protection against challenge infection [108,165,166], possibly indicating that the underlying immune response (i.e., the polarisation of induced responses as well as the cellular source of IFNγ) might be an important factor in assessing the potential outcome.

Elucidating the cellular and molecular details of the complex T cell responses against ASFV infections requires more dedicated research but will ultimately facilitate a more comprehensive understanding of the pathogen–host interactions during ASFV infections, hopefully leading to the design of efficient vaccines against this fatal viral disease.

## Figures and Tables

**Figure 1 pathogens-11-00274-f001:**
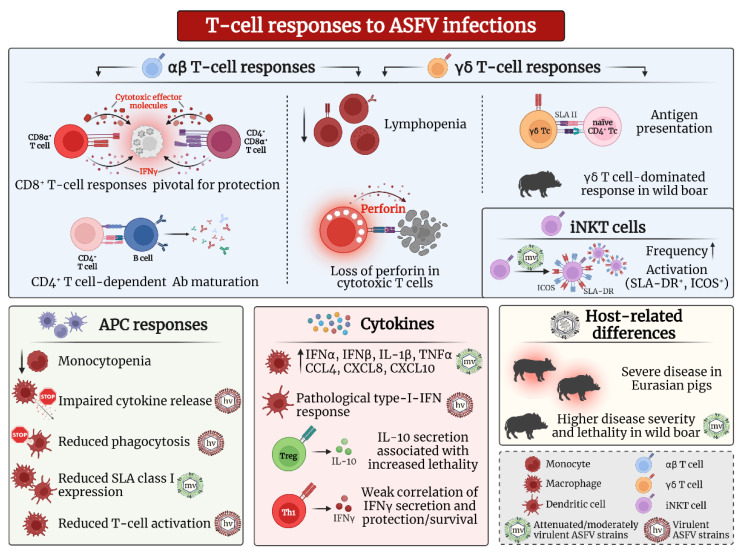
Adaptive cellular responses during ASFV infection. T cells have been shown to be particularly important for the survival of ASF infections. Antiviral T cell responses are predominantly executed by CD8α+ T cells and include cytotoxicity as well as secretion of proinflammatory cytokine-like IFNγ. The widespread loss of perforin in cytotoxic T cells potentially indicates consumption or a switch to other cytotoxic effector molecules. During ASFV infections, CD4+ Th cells support B cell responses and essential antibody (Ab) maturation. Studies on nonconventional T cells, such as γδ T cells and invariant Natural Killer T (iNKT) cells, point to a role in antigen presentation and possibly also as mediators of cytotoxic responses. The induction of adaptive responses, facilitated by professional antigen-presenting cells (APC), is also significantly disturbed during ASFV infections. Virulent ASFV strains often impair vital cellular functions, such as cytokine secretion and phagocytosis, or induce pathological cytokine responses, thereby diminishing targeted antiviral responses. Adaptive cytokines, such as anti-inflammatory IL-10 and proinflammatory IFNγ, are often found in moribund animals during likely irreversibly derailed responses and do not correlate well with protection. While both Eurasian suids, domestic pigs and wild boar, are highly susceptible to ASFV infections, experimental studies found higher disease severity and lethality in wild boar even during moderately virulent ASFV infections. Wild boar were also found to have a significant bias towards γδ T cell responses. Where applicable, effects of moderately virulent or attenuated ASFV strains were marked with a green virus (mv), and effects of highly virulent ASFV strains were marked with a red virus (hv). Created with BioRender.com (accessed on 17 February 2022).

## Data Availability

Not applicable.

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
