# Peer review of "Adaptive Cellular Immunity against African Swine Fever Virus Infections"

_pathogens, 2022, doi:10.3390/pathogens11020274_

Round 1
Reviewer 1 Report
Line 51 - maybe it should be mentioned here as well, that the clinical picture depends as well on the genotype of the ASFV, while genotype I and genotype II has completely different clinical signs.
Line 177 - the abbreviation of dendritic cells (DC) should be included firstly and then it should be used later on in the text.
Line 215 - check the spelling.
It is suggested to use the same terms all over the text - for example, "moderately virulent Estonia2014 infection", "moderately virulent Estonia2014 ASFV strain", "moderately virulent Estonia2014 ASFV infections", "moderately virulent Estonia 2014", while it makes a lot of confusions if the authors have the same ASF virus strain in mind.
Another example - LIne 390 - Moderately virulent ASFV “Estonia2018”, where the name of the strain is in brackets...
Same remarks as previously for usage same terms in the text for the Armenia2007 strain - "highly virulent Armenia2007 ASFV strain", "with Armenia2007", "highly virulent Armenia2007", "Armenia2007 infection".
Lines 547-549 - "Antibodies specific for ASFV found in hunted wild boar in Estonia suggested that at least some animals survive field infections [136], but this is usually not replicated in experimental infections [120]." - is it really only one ASF affected country that has reported the presents of antibodies in the hunted wild boars? The current sentence leads to confusion...
Author Response
Dear Reviewer 1
Please, find attached the revised version of our manuscript entitled “Adaptive cellular immunity against African swine fever virus infections” by Alexander Schäfer, Giulia Franzoni, Christopher L. Netherton, Luise Hartmann, Sandra Blome, and Ulrike Blohm for completion of the review process.
We have read your report carefully and thank you for the helpful comments and critique.
We have standardized the names of the ASFV strains and hope to have eliminated irritations in the readability. The abbreviation DC was already introduced at the beginning of the chapter, but as suggested, we have explained it again in the subchapter DC to increase readability here as well. In addition, we would like to respond to the remaining two points as follows:
Line 51 - maybe it should be mentioned here as well, that the clinical picture depends as well on the genotype of the ASFV, while genotype I and genotype II has completely different clinical signs.
We respectfully disagree on this point. There is no evidence that B646L/p72 genotyping predicts any ASFV biology. The 400 base pairs that encode the C-terminus of the major capsid protein of ASFV represents a useful tool for molecular epidemiology studies in areas with diverse circulating strains of ASFV, i,e. Southern and Eastern Africa. However, you cannot use to the genotype of a given virus to make predictions about clinical signs after infection. For example, the Lisbon 57, Lisbon 60, Benin 97/1 and OUR T88/1 isolates are highly virulent and kill animals within 4 to 10 days with clinical signs typical of ASF. The Malta 1978 strain causes a more prolonged clinical picture, killing approximately 50% of the animals it infects and hence was classified as moderately virulent. NHP/68, OUR T88/3 and the two new strains isolated in China all cause limited clinical signs or chronic disease. All of these viruses are genotype I. A similar range of clinical signs can be observed in genotype II viruses, i.e. Georgia 2007/1 is highly virulent, Estonia 2014 is moderately virulent and Latvia 2017 causes chronic disease. The sentence as written is correct, the clinical picture is dependent on the strain of virus that infects the animal and is independent of the genotype.
Lines 547-549 - "Antibodies specific for ASFV found in hunted wild boar in Estonia suggested that at least some animals survive field infections [136], but this is usually not replicated in experimental infections [120]." - is it really only one ASF affected country that has reported the presents of antibodies in the hunted wild boars? The current sentence leads to confusion...
We have generalized the statement to "Eastern European countries and Sardinia." In fact, there are repeated reports of ASFV seropositive wild boars in the hunting population. However, publications on this are rare, but we have added literature references expanding this point. We have also clarified the sentence regarding experimental infections.
Kind regards,
Ulrike Blohm
Reviewer 2 Report
The present paper by Alexander Schäfer et al. reviewed the adaptive cell-mediated immune responses to African swine fever virus (ASFV) infection. Generally, the paper was well organized and presented. Several concerns should be addressed prior to acceptance of publication.
- A nice review paper usually includes self-created tables and/or figures to summarize and integrate the key findings in this field.
- Insights and suggestions on the future research priorities should be provided in the Concluding Remarks.
- The reference styling should be consistent based on the requirements of the journal.

Author Response
Dear Reviewer 2
Please, find attached the revised version of our manuscript entitled “Adaptive cellular immunity against African swine fever virus infections” by Alexander Schäfer, Giulia Franzoni, Christopher L. Netherton, Luise Hartmann, Sandra Blome, and Ulrike Blohm for completion of the review process.
We have read your report carefully and thank you for the helpful comments and critique. We would like to respond to your points as follows:
1.A nice review paper usually includes self-created tables and/or figures to summarize and integrate the key findings in this field.
Thanks for the note. We agree, that it is useful to have a summary illustration in a review. We designed a figure that summarizes the current knowledge about the cellular immune response against ASFV infection and added it to the concluding chapter.
2.Insights and suggestions on the future research priorities should be provided in the Concluding Remarks.
First, we decided to address open questions at the end of each chapter. Based on your suggestion, we have rearranged the manuscript and summarized open questions and the outlook for various further experiments in a longer concluding chapter.
3.The reference styling should be consistent based on the requirements of the journal.
Thank you, we checked and changed the reference styling.
Kind regards,
Ulrike Blohm
Reviewer 3 Report
I reviewed the paper entitled “Adaptive cellular immunity against African swine fever virus infections”. In this review authors describe multiple cells involved in the immune response of ASFV.
Overall, I consider this manuscript as an excellent review, well written by authors with a large experience in the topic. I don’t have any concern to recommend the current version of this manuscript for publication.
A small suggestion to improve this current version would be the inclusion in the section of macrophages a small discussion about the paper published by Zhu et al., 2019, regarding the mechanism of pathogenesis described in ASFV based on microarray analysis in macrophages. Also, I consider important to highlight the importance of macrophages in the evaluation of recombinant viruses carrying some gene deletions, since the ability to grow of these viruses in macrophages (when compared with the WT), is in many cases a good predictor of the virulence of these viruses in pigs (see multiple studies from Borca et al).
Zhu JJ, Ramanathan P, Bishop EA, O'Donnell V, Gladue DP, Borca MV. Mechanisms of African swine fever virus pathogenesis and immune evasion inferred from gene expression changes in infected swine macrophages. PLoS One. 2019;14(11):e0223955. Published 2019 Nov 14. doi:10.1371/journal.pone.0223955
Author Response
Dear Reviewer 3
Please, find attached the revised version of our manuscript entitled “Adaptive cellular immunity against African swine fever virus infections” by Alexander Schäfer, Giulia Franzoni, Christopher L. Netherton, Luise Hartmann, Sandra Blome, and Ulrike Blohm for completion of the review process.
We have read your report carefully and thank you for the praise and helpful comments. We would like to respond to your points as follows:
A small suggestion to improve this current version would be the inclusion in the section of macrophages a small discussion about the paper published by Zhu et al., 2019, regarding the mechanism of pathogenesis described in ASFV based on microarray analysis in macrophages.
Thank you for the comment, we have added a paragraph with a discussion of the recommended paper.
Also, I consider important to highlight the importance of macrophages in the evaluation of recombinant viruses carrying some gene deletions, since the ability to grow of these viruses in macrophages (when compared with the WT), is in many cases a good predictor of the virulence of these viruses in pigs (see multiple studies from Borca et al).
While we agree that recombinant viruses that replicate poorly in macrophages are likely to be attenuated in pigs (e.g. Georgia I177L deletion, Pr94/6 9GL deletion), there are many examples of recombinant ASFV and field strains that replicate well in macrophages and are attenuated in pigs. Many of the current vaccine candidates replicate perfectly well in primary macrophage cultures (e.g. the candidates from Harbin and the recent publications from Pirbright). In addition, NHP/68 and OUR T88/3 replicate well in macrophages and are attenuated in pigs when compared to pathogenic genotype I viruses. Therefore, we would like to refrain from making a general statement in this regard.
Kind regards,
Ulrike Blohm
Round 2
Reviewer 2 Report
My major concerns have been addressed adequately and the revised manuscript is now acceptable.